# Tipping points in coupled human-environment system models: a review

Isaiah Farahbakhsh[1], Chris T. Bauch[2], Madhur Anand[1]

[1]School of Environmental Sciences, University of Guelph, Guelph, N1G 2W1, Canada
[2]Department of Applied Mathematics, University of Waterloo, Waterloo, N2L 3G1, Canada

*Correspondence to*: Madhur Anand (manand@uoguelph.ca)

**Abstract.** Mathematical models that couple human behavior to environmental processes can offer valuable insights into how human behavior affects various types of ecological, climate, and epidemiological systems. This review focuses on human drivers of tipping events in coupled human-environment systems where changes to the human system can lead abruptly to desirable or undesirable new human-environment states. We use snowball sampling from relevant search terms to review the modeling of social processes–such as social norms and rates of social change–that are shown to drive tipping events, finding that many affect the coupled system depending on the system type and initial conditions. For example, tipping points can manifest very differently in human-extraction versus human-emission systems. Some potential interventions, such as reducing costs associated with sustainable behavior, have intuitive results. However, their beneficial outcomes via less obvious tipping events are highlighted. Of the models reviewed, we found that greater structural complexity can be associated with increased potential for tipping events. We review generic and state-of-the-art techniques in early warning signals of tipping events and identify significant opportunities to utilize digital social data to look for such signals. We conclude with an outline of challenges and promising future directions specific to furthering our understanding and informing policy that promotes sustainability within coupled human-environment systems.

**Non-technical summary.** Mathematical models that include interactions between humans and the environment can provide valuable information to further our understanding of tipping points. Many social processes such as social norms and rates of social change can affect these tipping points in ways that are often specific to the system being modeled. Higher complexity of social structure can increase the likelihood of these transitions. We discuss how data is used to predict tipping events across many coupled systems.

## 1 Introduction to tipping points in coupled human-environment systems models

Humans are facing environmental catastrophes of their own making, like climate change and biodiversity declines, at local and global scales, and yet avoiding these catastrophes still poses complex challenges for sustainable behavior and policy interventions (Steffen et al., 2017). Traditionally, mathematical models of environmental

systems have represented human impacts through fixed, static parameters or functions independent of the environment's current state (Binford et al., 1987; Bosch, 1971; Chaudhuri, 1986; Getz, 1980), and these models can be useful to inform optimal levels of sustainable extraction for short timescales. However, for longer timescales, where human dynamics can evolve, it may be necessary to include human behavior endemically in the modeling framework to allow for human-environment feedback to occur (Bauch et al., 2016; Innes et al., 2013; Lade et al., 2013; Schlüter et al., 2012). Coupled human-environment system (CHES) models combine environmental (e.g., ecological, epidemiological, and climate) models with human behavior and population dynamics (Bury et al., 2019; Carpenter et al., 2009; Farahbakhsh et al., 2022; Innes et al., 2013; Lade et al., 2013; Phillips et al., 2020; Sethi and Somanathan, 1996). For example, in Innes (2013), the amount of forest cover influences the proportion of the population that conserves forest ecosystems. The influence of each subsystem on one another often occurs as two-way (positive and/or negative) feedback loops. In a positive (self-reinforcing) feedback loop, variable 'A' causes an increase in variable 'B' which then causes an increase in 'A'. In a negative feedback loop, 'A' causes an increase (respectively, decrease) in 'B' which causes a decrease (respectively, increase) in 'A'. The inclusion of these feedbacks leads to increased diversity in the qualitative behavior of the system, such as whether the long-term dynamics converge to a sustainable or depleted environmental state, or cycle over time. Negative feedback promotes a return to equilibrium (Figure 2a) and can increase the system's capacity to respond to disturbances and adapt in ways that allow the system to maintain the function of social and ecosystem services, which is sometimes referred to as "resilience" (Folke, 2006).

Human-environment negative feedback loops via processes such as public concern pressuring governments to introduce environmental legislation can be powerful and there are many historical examples of it occurring (Dunlap, 2014; Grier, 1982; Mather and Fairbairn, 2000; Stadelmann-Steffen et al., 2021). Forest cover in Switzerland doubled, following an all-time low in the first half of the 19th century. This was brought about by public concern responding to food shortages and floods, which triggered local regulation, the formation of the Swiss Forestry Society, and the first federal forestry law enacted in 1876 (Mather and Fairbairn, 2000). Similarly, the bald eagle population in North America recovered significantly after the banning of DDT by the EPA in 1972. This was instigated by public outcry following the publication of Rachel Carson's *A Silent Spring* in 1962 which linked DDT in the environment to low reproduction of birds and their declining population (Dunlap, 2014; Grier, 1982). In both cases, the gradual recovery of the population was not brought about simply by governmental legislation. There were strong movements in the public and scientific spheres, directly responding to perceived environmental risk which pressured governing bodies to enact immediate reform (Dunlap, 2014; Grier, 1982; Mather and Fairbairn, 2000). We interpret these two examples as negative feedback loops in a coupled human-environment system because a decline in forest/eagle abundance stimulated a response by humans which led to the recovery of the environmental system (Figure 2a). These negative feedback loops are pervasive in the CHES models that we examine here.

The historical examples above describe negative feedbacks promoting a return to a single environmentally beneficial equilibrium; however, in many cases, this does not happen and the system can persist in a depleted state. For example, the desertification of regions once rich in vegetation could become a positive feedback loop maintaining the new desert state (Hopcroft and Valdes, 2021; Pausata et al., 2020). When systems can persist in qualitatively different states (also referred to as "regimes"), we say that they exhibit alternative stable states (May, 1977; Lenton et al., 2008, Henderson et al. 2016). In mathematical models, alternative stable states are self-reinforcing for a range of parameters, for example, low harvest rates can promote a state of high biomass and high harvest rates can promote a state of low biomass in many extractive CHES (Farahbakhsh et al., 2021; Henderson et al., 2016; Richter and Dakos, 2015; Richter et al., 2013; Schlüter et al., 2016). Tipping points refer to critical points on this boundary between two alternative stable states. Near this boundary, small perturbations can be amplified through nonlinear self-reinforcing positive feedback loops. This leads to a qualitatively different system state and characteristic behavior, known as a "regime shift", in a relatively short amount of time. When the system has entered a new regime, there are often positive or negative feedback loops that make it difficult to reverse this change. This self-perpetuating nature of some initial change through nonlinear feedbacks leading to qualitative and often long-term system change is a universal characteristic of many commonly studied tipping points. In many cases, a return to the system's previous state can be more difficult than anticipated, requiring additional effort rather than merely a return to parameters before the tipping point, a phenomenon known as hysteresis, which can make mitigation and adaptation efforts challenging. Systems near a tipping point can exhibit (often abrupt) regime shifts through gradual changes or noise in forcing parameters, which is a main focus of much of the bifurcation theory literature (Figure 1a, Box 1.1). The scope of models presented in this review will not include other types of tipping points such as those caused by a short sharp shock (s-tipping, or shock-tipping, where the system does not have to exist near this point for a regime shift to occur) (Figure 1b) (Boettiger and Batt, 2020; Halekotte and Feudel, 2020) or "rate-induced tipping", which is a distinct phenomenon induced by the rate of change of parameters (Ashwin et al., 2012). Tipping events describe the crossing of a tipping point and can be used interchangeably with regime shifts.

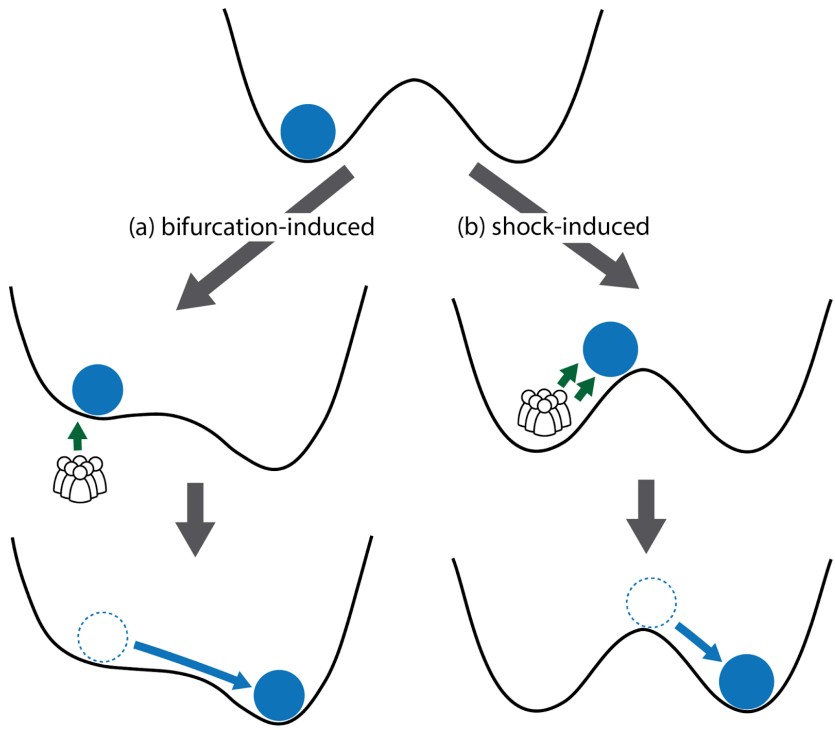

**Figure 1: Two types of tipping events; bifurcation-induced tipping (a), where the drivers are gradual changes to system parameters leading to a tipping event, and shock-induced tipping (b), where a perturbation to the system causes it to enter an alternative stable state through the crossing of a tipping point. Many social tipping points are caused by a combination of both types of tipping events. The blue circle represents the current state of the system.**

Bifurcation theory has been applied to study tipping points in a vast number of environmental models (May and
Oster, 1976; Brovkin et al., 1998; Ghil and Tavantzis, 1983; Wollkind et al., 1988); however, more recently,
researchers have identified abrupt shifts in environmental systems for which bifurcation theory has yet to be
explicitly applied (Dakos et al., 2019; Lenton, 2020, 2013). For example, during the mid–Holocene, the Sahara was
much more humid than at present, showing evidence of shrub and savannah biomes as well as the expansion of
lakes, an alternative stable state to what we know as its current desert state. It is hypothesized that around 5,000
98 years ago, the gradual weakening of the North African Monsoon led to an abrupt decrease in vegetative cover, due to
99 positive feedback between reduced surface albedo and precipitation, bringing the Sahara into a stable desert state
(Hopcroft and Valdes, 2021; Pausata et al., 2020). In more dominantly human systems, many pivotal revolutions can
also be framed as tipping events where gradual changes are reinforced by positive feedback loops, leading to a new
political or technological stable state (Lenton et al., 2022). Social tipping also occurs in financial systems such as in
the 2008 financial crisis. Here, the bankruptcy of Lehman Brothers led to a rise in public panic around the stability
of markets, causing banks to increase their liquidity, amplifying the crisis in other economic sectors and leading to a

global recession (Van Nes et al., 2016). These are just two of many examples illustrating how important tipping

points are as a phenomenon, in both human and environmental systems, and coupling these systems using

mathematical models could lead to further insights.

Since the beginning of the Anthropocene and with our growing awareness of human impacts on the environment,

tipping points are increasingly being conceptualized within the context of coupled human-environment systems

(Bauch et al., 2016; Henderson et al., 2016; Lenton et al., 2022; Milkoreit et al., 2018). Tipping events can lead to

highly beneficial or catastrophic outcomes for humans, especially when an environmental change occurs in the

presence of social hysteresis. An example of detrimental tipping is in the forests of Kumaun and Garhwal in

Northern India, where, prior to British colonization, wood harvest was sustainably regulated through social norms

and strict rules enforced by local village councils. When the British colonial government imposed its own rules on

the use of forests, these social norms broke down. Eventually, protests led to British lumber restrictions being

removed, but the system subsequently experienced rapid deforestation rather than a return to its previous levels

under local management. Here, the social system crossed a tipping point between a self-organized common property

regime to one of open access devoid of self-regulating sanctions (Somanathan, 1991). This system has been modeled

using a dynamical systems approach that allows for a quantitative understanding of the human drivers leading to

these tipping events (Sethi and Somanathan, 1996). Contrasting this example, tipping events can also result in

environmental change that is beneficial to humans and the environment. The rapid response of the international

community to the hole in the ozone layer has been interpreted by some as an example of a CHES undergoing tipping

events caused by self-perpetuating change through political, technological, and behavioral forces

(Stadelmann-Steffen et al., 2021). In the 1970s, scientists demonstrated the detrimental effects of CFCs on the ozone

layer, which could be viewed as the initial driver of the following socio-climate tipping events. This led to public

concern, prompting several countries to ban the use of CFCs in aerosols. Through the enactment of national policies,

public awareness increased, leading to more public pressure for national and international policy change, an example

of a positive feedback loop. In parallel, these national bans of CFCs, especially in the US, led to the development of

CFC alternatives, which prompted industries that could develop them to lobby for international policy. Increased

public awareness also led to widespread shifts in social norms stigmatizing and boycotting the consumption of

CFCs, which further pressured industry to offer alternatives, another positive feedback loop. The interaction of

multiple tipping events at different scales led to the crossing of a global tipping point through the international

banning of CFCs, bringing an alternative stable state of very low CFC emissions globally. (Andersen et al., 2013;

Cook, 1990; Epstein et al., 2014; Haas, 1992; Stadelmann-Steffen et al., 2021).

Tipping events associated with social processes as described in the preceding paragraph can be conceptualized

through positive feedback loops that capture a self-reinforcing process. In the case of social norms, this

self-reinforcing process may correspond to peer pressure or conformism that reinforces the dominant opinion or

belief. Depending on whether pro- or anti-mitigation opinions are currently dominant, this could lead to hysteresis (Figure 2b). The negative feedback loop that might normally regulate the CHES to exist in a state of intermediate environmental health and public support for sustainability (Figure 2a) could be overpowered by the positive feedback of social norms, leading the population to a state where either sustainability (or anti-sustainability) is strongly entrenched. If the conditions governing social learning or social norms move beyond a tipping point, the population may flip between these two norms, or alternatively it may move into a regime where social norms are instead dominated by the negative feedback loop, causing the population to exist in an interior state of partial sustainability. As such, negative feedback and positive feedback may be characteristic of any CHES and should be systematically studied.

This review aims to deepen our understanding of human drivers of tipping events in CHES models by exploring three crucial topics: the feedback loops and interactions between the human and environmental systems, the structural characteristics of the human system that influence tipping points, and the identification of early warning signals within human systems. By "human drivers", we refer to the changes in social parameters that elicit these non-linear tipping responses in either the environment, human system, or both. However, we also discuss aspects of social structure that may be conducive to tipping points. As most of the models reviewed are informed by dynamical systems and bifurcation theory, we primarily focus on systems that exist near tipping points and cross them through gradual changes in these drivers. In the following sections we review CHES model literature found using Google Scholar with the keywords: 'human environment system' OR 'socio-ecological system' OR 'social ecological system' OR 'human ecological system' OR 'human natural system' combined with 'tipping' OR 'regime shift' OR 'bifurcation'. These results were filtered manually to include only dynamical models that showed clear tipping behavior. Additional literature was found through a snowball approach using references from the sources found in this search as well as papers referencing these sources (Wohlin, 2014). The findings in this review highlight commonalities between the CHES models surveyed; however, some trends may be a result of both the dynamical models chosen and the relatively low diversity and volume of these models. The body of this review is split into two parts; the first part synthesizes results from CHES models, organized into processes and structures that drive tipping behavior, and the second part introduces early warning signals describing how they can be used to predict tipping events.

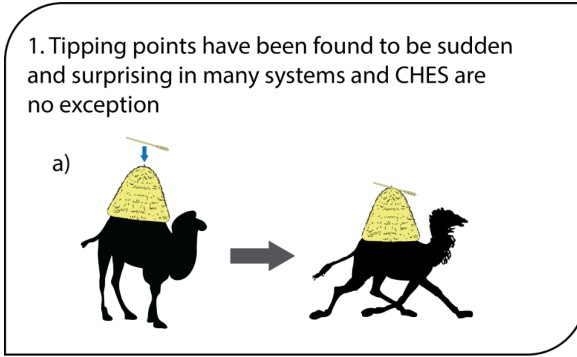

1. Tipping points have been found to be sudden and surprising in many systems and CHES are no exception

a)

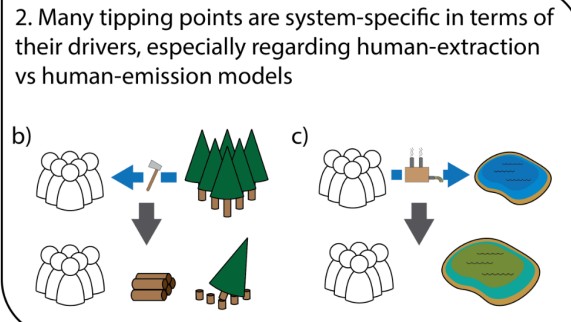

2. Many tipping points are system-specific in terms of their drivers, especially regarding human-extraction vs human-emission models

b)     c)

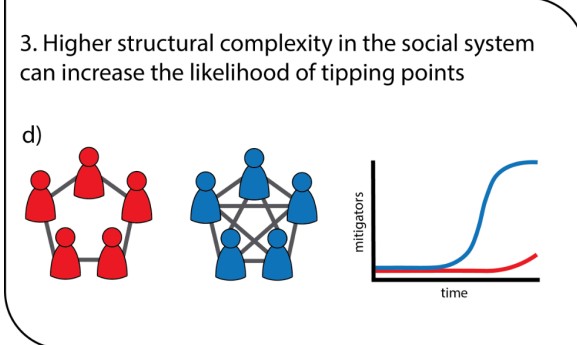

3. Higher structural complexity in the social system can increase the likelihood of tipping points

d)

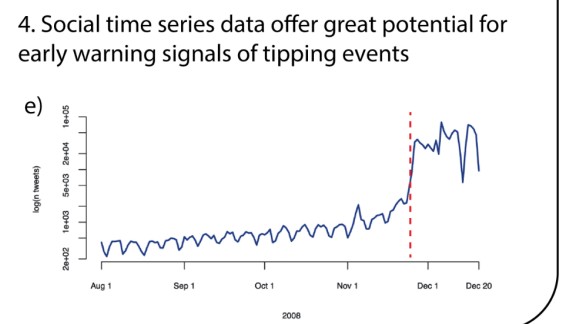

4. Social time series data offer great potential for early warning signals of tipping events

e)

**Box 1: Highlights of key findings from the synthesis of CHES models in this review. "The straw that broke the camel's back" illustrating bifurcation-induced tipping points (1a), in human-extraction systems (2b), increasing the speed of social change or the coupling strength leads to negative tipping points (i.e., ecological collapse), whereas in human-emission systems (2c), the effects of increasing the speed of social change or the coupling strength are model specific, higher connections in a social network leading to a positive tipping event, where the graph represents the proportion of mitigators in time (3d), time series data from Twitter showing an abrupt transition characteristic of a tipping event at the red dotted line (4e) from (Bollen et al., 2021).**

## 2 Processes and structures in human systems that cause tipping events in CHES models

In this section, we look at how social processes and structures cause tipping events. In order to have a better understanding of how these human drivers affect tipping, it is important to understand the basics of modeling human systems. Within CHES models, various factors, such as economic incentives, environmental considerations, and social pressures determine how individuals make decisions and interact with the environment. In most of the current modeling literature, individuals can choose between two behaviors (also referred to as opinions or strategies), one that is environmentally sustainable (also referred to as mitigation or cooperation) and another that is detrimental to the environment (also referred to as non-mitigation or defection). The perceived advantage of mitigation or non-mitigation relative to the current state of the human and environmental system can be quantified through a

"utility function". Common factors in the utility function are the rate of social learning, which determines the speed of human behavior change relative to environmental processes, social norms, which encourage the status quo or mitigation proportional to its frequency, cost of mitigation, which measures the economic cost of being a mitigator relative to a non-mitigator, and rarity-motivated valuation, which incentivizes mitigation as the environment approaches collapse (Bauch et al., 2016; Farahbakhsh et al., 2022; Tavoni et al., 2012). In most models that use social learning, individuals sample others in the population at a fixed rate and adopt a different behavior if the other behavior has a higher utility, with probability proportional to the difference in utility (Hofbauer and Sigmund, 1998; Schuster and Sigmund, 1983). This can also be formulated in a stochastic setting, where the probability of adopting a neighbor's behavior is a function of the difference in utility between behaviors (Schlag, 1998). Most of the models reviewed in this paper use social learning to represent human behavioral dynamics. There are also CHES models that do not include social learning such as Motesharrei (2014) and Dockstader (2019) where the human population is influenced by its current size and the state of the environment; however, these are outside the scope of this paper.

Many human behaviors, such as resource extraction and pollution, have direct detrimental impacts on the environment; however, the severity of these impacts is often hard to predict. In many CHES models, small changes in parameters governing human behavior and social processes can lead to the abrupt collapse of sustainable states through tipping events that can cascade between the human and environmental systems (Bauch et al., 2016; Lade et al., 2013; Richter and Dakos, 2015; Weitz et al., 2016). Additionally, structural elements of the human system (i.e. an individual's degree of choice, population diversity), as well as how the social system is organized (i.e. through social networks), can affect tipping. These heterogeneous model elements are often only accessible in agent-based models, where humans are represented as individual agents that follow a set of rules. CHES models do not always exhibit tipping points under realistic settings for the human system (Bury et al., 2019; Menard et al., 2021); however, in this review, we focus on models with tipping points.

## 2.1 Coupling strength

Coupling strength (how strongly the subsystems are coupled) can have a significant effect on the occurrence of tipping points in both systems, and the nature of these transitions often depends on whether systems are 'human-extraction' or 'human-emission' (Box 1.2). In human-extraction systems (Box 1.2b), humans extract from an environmental resource such as in forest and fishery models. Stronger coupling in human-extraction models often leads to negative environmental outcomes. A common social parameter representing the coupling strength in these systems is the extraction effort of humans, which when increased past a critical threshold, leads to abrupt environmental collapse (Farahbakhsh et al., 2021; Richter and Dakos, 2015; Richter et al., 2013; Schlüter et al., 2016). For human-emission systems (Box 1.2c), where human activity increases levels of harmful outputs, such as pollution and climate models, coupling strength is instead represented by pollution rates. The influence of this coupling is less intuitive in human-emission systems, for example, in lake eutrophication models as the pollution of

mitigators is decreased, pollution levels also decrease until a threshold is reached, heralding a detrimental tipping point where mitigation collapses and pollution then reaches a high level (Iwasa et al., 2010, 2007). This occurs because when the lake water is not very polluted, there is less incentive to be a mitigator and high-polluting behavior becomes a new norm. It is important to note that these models do not account for individuals valuing the environment in a healthy state, for example through the centering of ecosystem services, and the above example may be an artifact of this assumption. There is a need to shift both our relationship to the environment as well as the assumptions in our models so that inherent value in environmental systems is central in any decision-making, even when the environment is far from collapse. This fundamental valuing of the environment is present in many traditional indigenous belief systems, where relationships to the local natural environment are incorporated and prioritized in all aspects of life (Appiah-Opoku, 2007; Bavikatte and Bennett, 2015; Beckford et al., 2010; McMillan and Prosper, 2016).

## 2.2 Rarity-motivated valuation

Rarity-motivated valuation represents the extent to which humans increase their mitigative behavior in response to the environmental variable (e.g., forest cover, endangered species population size) nearing a depleted state. In CHES models, this sensitivity of human response to the abundance of the natural resource/population is represented by a 'sensitivity' parameter and there are often two critical thresholds in the sensitivity parameter that lead to tipping. Increasing the sensitivity parameter beyond the lower threshold induces a tipping point from a depleted to sustainable environmental equilibrium (Ali et al., 2015; Barlow et al., 2014; Bauch et al., 2016; Drechsler and Surun, 2018; Henderson et al., 2016; Lin and Weitz, 2019; Sun and Hilker, 2020; Thampi et al., 2018; Weitz et al., 2016). The second threshold exists at high values of the sensitivity parameter, which may be counterintuitive, as one might expect high sensitivity to resource depletion to lead to more sustainable outcomes. In this case, the sustainable equilibrium is destabilized by overshoot dynamics or a state of chaos in both the human and environmental systems. These dynamics are caused by the human system being too sensitive to changes in the environment, leading to extreme oscillations in both human behavior and the environment, which increases the likelihood of collapse in mitigation and the state of the environment (Bauch et al., 2016; Henderson et al., 2016).

Rarity-motivated valuation can also be represented by a threshold in the state of the environment, below which humans shift towards sustainable behavior. In a common-pool resource model, lowering this threshold led to a series of tipping points that surprisingly resulted in a higher biomass equilibrium, although the trajectory to this state comes close to environmental collapse. This is in contrast to a high threshold, which leads to lower final biomass; however, the trajectory remains much farther from a depleted environmental state (Mathias et al., 2020). Similarly to high coupling in pollution models, one should be very careful to not interpret these results as stating "too much conservation is detrimental to the environment". They rest on model assumptions of a reactionary conservation paradigm, where there is less value in conserving when the environment is in a healthy state.

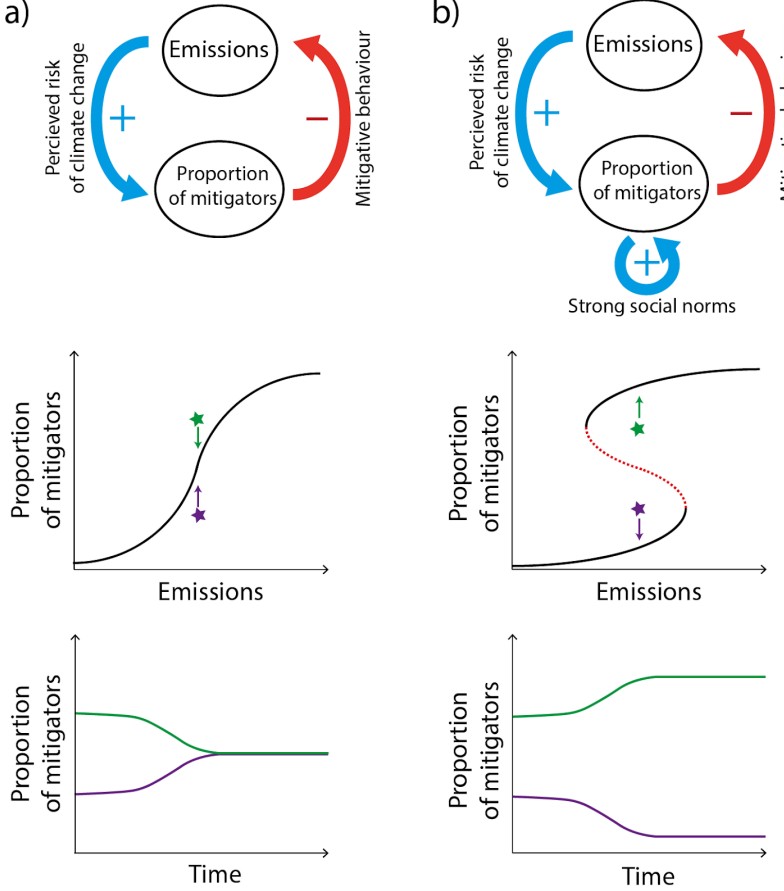

**Figure 2: Negative feedback between the human and environmental subsystems, supports convergence to the same equilibrium regardless of initial conditions (a). With strong majority-enforcing social norms, encouraging either mitigative or harmful behavior adds a positive feedback loop which makes the coupled system highly dependent on initial conditions (b). The top row shows the negative feedback loop between emissions and the proportion of mitigators, where (b) also includes the positive feedback of majority-enforcing social norms. In the middle row, equilibrium curves are plotted as a function of the maximum emissions of non-mitigators. Black solid lines represent stable equilibria and the red dotted line represents unstable equilibria. The green and purple curves in the bottom row are the trajectories for initial mitigation support and emission value given by the stars of the corresponding color in the upper row.**

277 **2.3 Social norms**

278 Introducing social norms can lead to alternative stable states and thus tipping points (Figure 2b), although the system

279 dynamics are highly dependent on both the type of social norms and initial conditions. Social norms are informal

rules emerging through social interaction that promote and discourage certain behaviors, especially around how humans relate to one another and the environment (Chung and Rimal, 2016). In models of small groups such as a community of fishers, they are often (rightly) assumed to support mitigative behavior by punishing those who violate norms by over-harvesting (Ostrom, 2000). However, at larger population scales, social norms can support either pro- or anti-mitigation behavior, on account of factors such as the politicization of actions relating to environmental, climate, and public health crises (Stoll-Kleemann et al., 2001; Van Boven et al., 2018; Latkin et al., 2022). Unlike a fisher in a small community, for instance, a climate denier may not acknowledge themselves as a 'defector' who is harming a public good, but rather view the climate activist as 'defecting' against a free society. Thereby, social norms have the ability to encourage behavior that is harmful to both human and environmental well-being, over larger spatial and temporal scales (Bury et al., 2019; Latkin et al., 2022; Menard et al., 2021; Stoll-Kleemann et al., 2001; Van Boven et al., 2018).

Social norms can be represented as majority-enforcing, incentivizing the behavior of the majority, or mitigation-enforcing, such as sanctions, which only incentivize mitigation, relative to the proportion of mitigators in the current state of the system. In CHES models, increasing the strength of majority-enforcing norms leads to an increased number of regimes as well as bistable (two stable states) regimes (Figure 2b), made up of a single dominant behavior, which is highly dependent on the initial proportion of behaviors in a population (Ali et al., 2015; Barlow et al., 2014; Bauch et al., 2016; Bury et al., 2019; Phillips et al., 2020; Sigdel et al., 2017; Thampi et al., 2018). This occurs because these norms are indifferent to the type of behavior they enforce (i.e. sustainable vs harmful actions), and they act as a double-edged sword that reinforces the status quo through a positive feedback loop, where the dominant behavior becomes more prevalent (Figure 2b). On the other hand, increasing mitigation-enforcing social norms lead to a transition of the environmental system into a sustainable equilibrium (Chen and Szolnoki, 2018; Iwasa et al., 2010; Lafuite et al., 2017; Moore et al., 2022; Schlüter et al., 2016; Tavoni et al., 2012), sometimes through an intermediate regime of oscillatory dynamics (Iwasa et al., 2007). In a lake pollution model, along with decreasing the likelihood of environmental collapse, this increase in mitigation-enforcing social norms also led to the appearance of alternate stable states (Sun and Hilker, 2020). These findings show that stronger social norms lead to a greater number of tipping points; however, the trajectories brought about by these tipping points are highly dependent on the type of social norms (mitigation- or majority-enforcing) as well as the current dominant social behavior.

## 2.4 Cost of mitigation

Reducing the cost of mitigation often leads to beneficial tipping points; however, these tipping points can depend on the rate of social change as well as social norms. Although it is intuitive that reducing costs or increasing economic incentives associated with mitigative action will have beneficial impacts on the environment, CHES models also show that this beneficial change can occur through tipping points (Bauch et al., 2016; Drechsler and Surun, 2018;

Milne et al., 2021; Moore et al., 2022; Sigdel et al., 2017; Thampi et al., 2018). In coupled social-epidemiological models, where the environmental state is the proportion of infected individuals, mitigation cost is represented through the economic cost or perceived risk of vaccination. Decreasing this cost leads to beneficial tipping points from a state with low pro-vaccine opinion and vaccine coverage to high pro-vaccine opinion and vaccine coverage (Phillips et al., 2020). Conversely, increasing this cost leads to a state of high infection and low vaccination. This detrimental tipping point occurs in the human system at lower levels of vaccination cost when majority-enforcing social norms are low, leading to widespread anti-vaccine opinion before the infection becomes endemic again (Phillips and Bauch, 2021). Decreasing profits of individuals engaging in non-mitigative behavior can also lead to an abrupt shift to a state of pure mitigators (Shao et al., 2019; Wiedermann et al., 2015); however, this transition can be dependent on a low rate of social change (Wiedermann et al., 2015). Other models demonstrate tipping in the other direction where increasing non-mitigators' payoff brings about a regime shift to pure non-mitigation and environmental collapse (Richter et al., 2013; Tavoni et al., 2012). Similarly, a common-pool resource model that uses machine learning in a continuous strategy space shows tipping to a depleted resource regime when the costs associated with harvesting are too low (Osten et al., 2017). An analog to mitigation cost is taxation rates, which resource users pay towards public infrastructure mediating resource extraction. In a model where individuals can choose to work outside of the system, pushing taxation rates to high or low levels tips a sustainable regime where institutions are at full or partial capacity to a collapse of institutions (Muneepeerakul and Anderies, 2020). In another model, only individuals with high extractive effort are subject to taxation, and increasing this taxation rate brings about a beneficial tipping point to a sustainable regime. However, the size of this sustainable region in the parameter space is smaller with multiple governance nodes evolving through social learning compared to a single taxing entity (Geier et al., 2019). However the cost of mitigation is represented, increasing the relative economic incentive of mitigation behavior has the potential to bring about beneficial tipping to a sustainable regime.

## 2.5 Rates of social change and time horizons

Human and environmental change often occur on different timescales and their relative rates of change play a major role in the long-term dynamics of the coupled system and whether or not tipping points will occur. Increasing the rate of social change (in most cases, social learning) leads to collapse in human-extraction models due to overshoot dynamics, whereas, in human-emission models, the impacts of the rate of social change are more model-specific. In both types of models, increasing the time horizon in decision-making is beneficial. In CHES models, these rates of change can be controlled by the rate of social learning which determines how frequently individuals interact and consequently, the pace of behavioral change within a population. Changes in the speed of the human system can have very different outcomes depending on the nature of human-environment coupling (Box 1.2). In human-extraction models, increasing the speed of the human system relative to the environment often destabilizes sustainable equilibria, leading to oscillations in both systems and, in many cases, the abrupt collapse of the environmental system. These overshoot dynamics occur as humans change their behavior too quickly to allow for

the environment to stabilize. On the other hand, decreasing the relative speed of human dynamics usually brings about beneficial tipping events leading to a state of high forest cover (Figueiredo and Pereira, 2011), and supporting mitigators for a generalized resource (Hauert et al., 2019; Shao et al., 2019). These beneficial effects have also been observed in adaptive network models where individuals imitate their neighbors depending on the profitability of their strategies. In these models, the reduced speed of social change leads to beneficial outcomes as the resource is allowed more time to stabilize as decisions regarding extractive levels occur (Barfuss et al., 2017; Geier et al., 2019; Wiedermann et al., 2015). Other relative rates of change can also significantly influence the existence of a sustainable regime. For example, in an agricultural land use model, increasing the speed of agricultural expansion and intensification relative to human population growth leads to the collapse of both the natural land cover and human population (Bengochea Paz et al., 2022).

In human-emission models, increasing the speed of social interaction is more model-specific. In some cases, such as forest-pest and climate systems, increasing the speed of the human system leads to better mitigation of environmental harms in the short term. However, long-term sustainability often requires additional social interventions such as reducing mitigation costs and increasing levels of environmental concern (Ali et al., 2015; Barlow et al., 2014; Bury et al., 2019). In lake pollution models, higher relative speeds of social dynamics can destabilize low-pollution equilibria, leading to oscillations and eventually a polluted state with no mitigation (Iwasa et al., 2010, 2007; Sun and Hilker, 2020). This is a similar phenomenon to the overshoot dynamics that occur when the human system is extremely reactive to the environment discussed in the case of rarity-motivated valuation; however, these outcomes are highly dependent on other social parameters. In a related model, with no social hysteresis, represented by mitigation-enforcing social norms, and strong environmental hysteresis, represented by a high phosphorus turnover rate, fast social dynamics could stabilize oscillations, leading to a low-pollution equilibrium (Suzuki and Iwasa, 2009). The emergence of oscillations under low rates of social learning, which was not observed in similar models is likely due to the environmental system being in a bistable state under strong hysteresis, such that even slow changes in the human system could tip the lake system to an alternative stable state.

When looking at relative rates of change in human and environmental systems, it is clear that the pace of the human system can be more readily influenced by interventions. This suggests an urgent need to further study the relationship between social and ecological timescales across a wide range of coupled systems to aid in sustainable policy-making decisions (Barfuss et al., 2017). Additionally in many models, the length of time horizons that humans take into account when deciding how they interact with the environment has a significant beneficial effect on conserving natural states and mitigating harmful action (Barfuss et al., 2020; Bury et al., 2019; Henderson et al., 2016; Lindkvist et al., 2017; Müller et al., 2021; Satake et al., 2007). A high degree of foresight in decision-making is a fundamental basis for many indigenous belief systems across the world. One manner in which this shows up is

393 in land stewardship where care for the environment is prioritized as a means to ensure the health of many
generations in the future (Appiah-Opoku, 2007; Beckford et al., 2010; Ratima et al., 2019).

**2.6 Social traits**

The inclusion and distribution of traits within agents can play a large role in determining the occurrence and types of
tipping points within the coupled system, where increasing the modeled heterogeneity in social traits can lead to
more tipping and also promote sustainable outcomes (Box 1.3). The majority of models discussed in the previous
section only allow humans to choose between two strategies; mitigation and non-mitigation. The inclusion of
additional strategies, determining how individuals interact with the environment and each other, can alter the
potential for tipping points. For example, a common-pool resource model included a third strategy of conditional
mitigation (Richter and Grasman, 2013). Under this additional strategy, agents act as mitigators until the number of
non-mitigators reaches a certain threshold, where they then shift their behavior to non-mitigation. The addition of
this third strategy alters tipping dynamics in opposite ways, depending on the value of maximum harvesting efforts.
When efforts are high, the system is less prone to tipping; however, when they are low, tipping points are more
likely to occur. This third strategy also affects tipping points by masking internal social dynamics, leading to more
abrupt transitions, even when the system appears to be stable. This occurs when mitigators gradually change their
strategy to conditional mitigators which can go unnoticed as their interaction with the environmental system does
not change. However, when non-mitigation reaches high enough levels, there is a cascade of conditional mitigators
choosing non-mitigation, in an example of herd behavior, which puts abrupt harvesting pressure on the resource.
Another three-strategy model, where agents are partitioned by resource extraction rates, contrasts dynamics with and
without the trait of environmental concern (Mathias et al., 2020). In the absence of this trait, the human system
either tips to a state of high-extraction or low-extraction behavior, triggering either a detrimental or beneficial
environmental tipping point, respectively. Including environmental concern leads to an increased number of
cascading tipping points between both human and environmental systems. In a coupled agricultural model, where
human traits include management strategies that respond to socio-economic and climate conditions, decreasing the
diversity of these traits among agents in the system transitions the system from a sustainable state with high food
production, landscape aesthetics, and habitat protection to a state with low habitat protection (Grêt-Regamey et al.,
2019). As there are relatively few models that explicitly compare the complexity of social traits and their effect on
tipping points, it is difficult to say with certainty whether higher complexity will increase the likelihood of tipping
points in all CHES and whether this is due to a higher dimensionality of the system. However, the commonalities
between models showing the effects of social trait complexity are worth highlighting and will be put to the test with
future CHES models and empirical work.

**2.7 Social networks**

In many agent-based CHES models, individuals are structured on a social network, where they interact with others whom they share a link with. These models demonstrate how a higher number of connections in social networks increases the potential for tipping points, often through the emergence and growth of bistable regimes (Holstein et al., 2021; Sugiarto et al., 2015, 2017a) (Box 1.3). Additionally, the distributions of these connections play an important role. For example, in networks with the same average number of connections, higher heterogeneity of connections among nodes leads to tipping points occurring earlier under certain social (Ising model) dynamics (Reisinger et al., 2022). The distribution of resources in human-environment networks also affects the potential for abrupt environmental collapse. This often occurs in CHES network models where both human and environmental dynamics occur on a multi-layer network, representing partitioned or private resources. Resource heterogeneity can be controlled through the distribution of carrying capacities or the amount of resource flow between nodes in the network, where higher flows lead to homogeneous resource distributions. In both cases, increasing this heterogeneity can tip the system to a state of low extraction and high sustainability. In one model, heterogeneity in carrying capacities increases the likelihood of sustainable harvesters extracting from a resource with a large capacity, which they can maintain at high levels (in contrast to non-sustainable harvesters who extract at a higher rate), eventually convincing neighboring nodes to imitate their strategy (Barfuss et al., 2017). In another model, heterogeneity through lower resource flows also leads to high-extraction nodes over-exploiting their resource and losing profits in the long run, de-incentivizing neighbors to imitate their behavior. Interestingly, optimal resource flow, which minimizes the likelihood of resource collapse is found to be close to the critical threshold of resource flow, above which the coupled system collapses. As optimal resource flow decreases the likelihood of collapse by supplementing resources harvested at high levels, this confers an advantage to high resource extraction. Increasing past optimal levels leads to similar resource levels among high and low-extraction nodes, resulting in higher profits from high-extraction nodes, incentivizing the entire human system to eventually choose the high-extraction strategy (Holstein et al., 2021).

Heterogeneity of human interaction can be quantified through homophily, the extent to which alike individuals interact. Homophily can play a large role in the occurrence and behavior of tipping points in CHES models occurring on social networks, often having a detrimental effect on the environmental system. In a common-pool resource model with two distinct communities, increasing segregation by lowering the probability that agents in separate communities will have a link, softens the abruptness of a single detrimental tipping point compared to when the communities are well-mixed. This is due to the occurrence of multiple intermediate tipping points within each segregated community; however, higher segregation adds more hysteresis to the system increasing the difficulty of reversing this transition and returning to a sustainable state (Sugiarto et al., 2017b). In a public goods game modeling climate change mitigation, where humans are partitioned into rich and poor agents, a transition to group achievement of mitigation goals occurs at a lower perceived risk when there is no homophily and agents are influenced by others from both economic classes equally (Vasconcelos et al., 2014). Another human-climate model

that included wealth inequality displayed an abrupt transition to lower peak temperature anomalies when homophily

between economic classes approached zero (Menard et al., 2021).

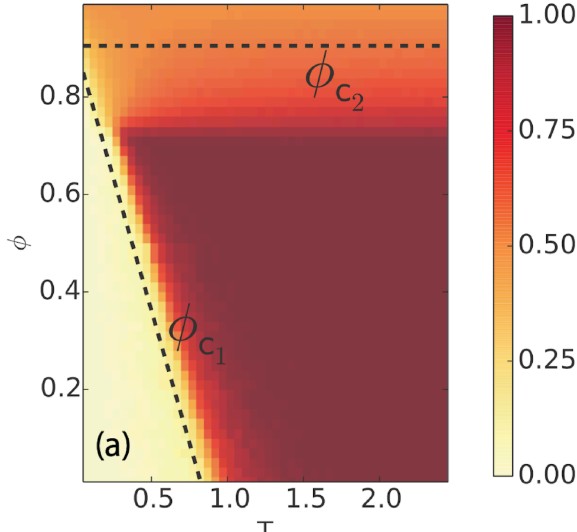

**Figure 3: Mean proportion of nodes that are mitigators for network model (a) and ODE model (b). $\phi$ is the rewiring probability and T is the time between social interactions. $\phi_{C_1}$ is the lower threshold and $\phi_{C_2}$ is the upper threshold, above which a fragmentation regime occurs. From (Wiedermann et al., 2015)**

Social networks are rarely static and their ability to evolve over time is represented in adaptive network models

where agents can break existing social links and create new ones, a process called "rewiring". Often this rewiring is

homophilic, meaning that agents are more likely to create a new social connection with others who share a similar

behavior. Common adaptive network CHES models have nodes representing renewable resource stocks with an

associated extraction level which can adopt a high extraction or low extraction level through imitating neighbors.

These models show that the level of homophilic rewiring can trigger regime shifts at both low and high levels,

where intermediate ranges correspond to a sustainable equilibrium. As agents can either choose to rewire or imitate

their neighbor, a low level of rewiring corresponds to a high speed of social interaction, which as discussed in

Section 2.5 can lead to detrimental tipping points. On the other hand, although high-rewiring leads to slower social

learning, it also brings about a fragmentation regime where social dynamics are dominated by homophily and the

network fragments into components based on strategy type, which makes widespread mitigation infeasible (Barfuss

et al., 2017; Geier et al., 2019; Wiedermann et al., 2015) (Figure 3). CHES models with social networks are still

relatively new and lack diversity in how they are formulated. For example, regarding the tipping points related to

rewiring social links, the lower threshold may be caused by increased social learning since in all models agents can

either rewire or imitate, but not both. There is still much to learn through isolating the effect of rewiring as well as
exploring a wide array of different model formulations of CHES on social networks.

## 3  Identifying early warning signals of tipping events in CHES

Although dynamical models can offer qualitative insight into potential trajectories of CHES resulting from specific
interventions, it is more difficult to use them to generate precise and reliable predictions. Given the potential for
severe environmental tipping points in the coming decades, it is extremely useful to be able to predict these abrupt
shifts without complete mechanistic knowledge of the system. The ability to predict tipping events with limited data
can allow policymakers to have more time preparing for future disasters, and given enough warning and political
will, an opportunity to avoid them or mitigate their severity. Rapidly growing research in early warning signals
(EWS) offers tools to monitor empirical time series data and warn of future tipping events that are likely to occur
(Bury et al., 2021; Dakos et al., 2012, 2015, 2008; Kéfi et al., 2014; Lapeyrolerie and Boettiger, 2021). Although
much of the work has been conducted on synthetic data, there are many studies that successfully predict historical
tipping events in both empirical human and environmental time series data such as the 1987 Black Monday financial
crash (Diks et al., 2019) as well as abrupt temperature shifts from paleoclimate datasets (Dakos et al., 2008).

### 3.1 Recent advances for detecting early warning signals

Much research has been done in the past few decades to develop tools for EWS using both empirical and synthetic
time series data (Bury et al., 2021; Dakos et al., 2012, 2015, 2008; Kéfi et al., 2014; Lapeyrolerie and Boettiger,
2021). Originally motivated by critical slowing down in bifurcation theory, where systems approaching a tipping
point show a slower recovery to equilibrium under perturbations, generic EWS measure trends in this "slowing
down" (Scheffer et al., 2009). The most commonly used methods compute the lag-1 autocorrelation and variance of
the residuals from detrended time series data. Other widely used methods involve metrics such as skewness,
measuring the asymmetry of fluctuations over time, and kurtosis, representing the likelihood of extreme values in
the time series data. A phenomenon known as flickering occurs when there is sufficient noise to rapidly force the
system between alternate stable states. In these cases, an increase in skewness and kurtosis is observed (Dakos et al.,
2012). As lag-1 autocorrelation does not account for correlation beyond a single time step, power spectrum analysis
has been used to look at changes in complete spectral properties, finding higher variations at low frequencies to
commonly occur before a tipping point (Dakos et al., 2012; Scheffer et al., 2009). In spatial systems, many EWS are
similar to those used in well-mixed systems, while also accounting for spatial variability. For example, Moran's I is
a spatial analog of lag-1 autocorrelation, which measures the correlation between neighboring nodes in a network
(Kéfi et al., 2014).

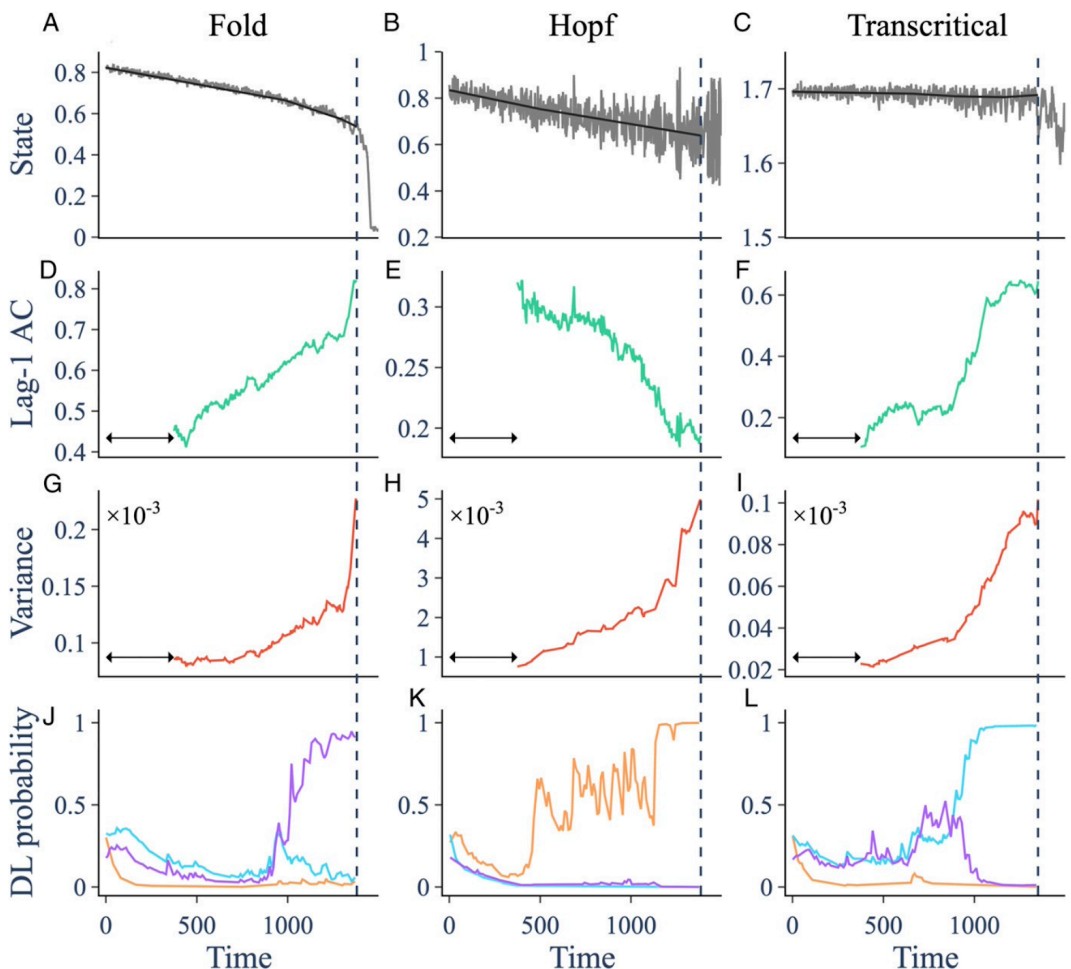

**Figure 4: Generic EWS (second and third row) as well as deep learning EWS (bottom row) for time series generated by two ecological models exhibiting different types of bifurcations (top row); fold (left), Hopf (middle), and transcritical (right). As well as being more reliable, deep learning EWS can also distinguish between the type of bifurcation being approached. In the bottom row, the DL algorithm gives probabilities for the occurrence of fold (purple), Hopf (orange), or transcritical (blue) bifurcations. Image taken from (Bury et al., 2021).**

Numerous spatial ecological systems exhibit patterns of patchiness preceding a tipping point. For example, in drylands, spotted vegetation patterns are hypothesized to be an EWS for the system approaching desertification (Kéfi et al., 2014). Coupled human-epidemiological models also show that spatial properties in the distribution of opinions on a social network offer potential EWS for the onset of disease outbreaks. Approaching this regime shift, the number of anti-vaccine clusters increases, and very close to the transition point, these communities coalesce into larger groups (Jentsch et al., 2018; Phillips et al., 2020). These clusters are quantified using a number of metrics, such as an increase in modularity as well as the mean number, size, and maximum size of communities and

pro-vaccine echo chambers (Phillips and Bauch, 2021). This is also in agreement with previous work done in percolation theory showing that phase transitions follow a breakup of connected components on the network (Newman, 2010).

One downside to the generic metrics discussed above is that they have the potential to fail in the presence of large amounts of noise where transitions can occur far from their analytically derived tipping point. A technique called dynamical network markers increases the dimensionality of the time series by transforming it from state variables to probability distributions of the mean and variance over a given window of time. This reduces the magnitude of noise in each dimension and in approaching a tipping point, one dominant group of variables will show a drastic increase in variance and correlation between other variables within that group. At the same time, the correlation between one variable in this dominant group and others outside the group will decrease. This technique has shown success with empirical data, such as predicting critical transitions in time series data for a eutrophic lake as well as the bankruptcy of Lehman Brothers (Liu et al., 2015), and flu outbreaks (Chen et al., 2019). Dynamical network markers have also been used on spatial systems such as those occurring on social networks through the use of hierarchical network representations. Here, networks are transformed into binary trees where leaves are the nodes from the original network and branches group nodes together at multiple resolutions. Through this hierarchical model, dynamical network markers use these multi-scale communities as the groups of variables that are analyzed (Li et al., 2023). This spatial technique offers a novel method for predicting tipping events for CHES using human data occurring on complex social networks.

A very recent addition to the EWS toolkit uses concepts from statistical physics such as average flux, entropy production, generalized free energy, and time irreversibility to predict tipping points in a shallow lake model much earlier than generic methods such as autocorrelation and variance, showing promise for use in real-time monitoring (Xu et al., 2023). Additionally, the field of machine learning has motivated data-driven approaches to EWS which do not explicitly make use of any statistical metrics in the time series data. Instead, deep learning algorithms are trained on large synthetic datasets using models that have and have not approached tipping points. In the majority of cases, these algorithms have performed significantly better at predicting tipping events than generic EWS indicators when tested on empirical datasets that exhibit abrupt transitions (Bury et al., 2021; Deb et al., 2022) (Figure 4). Deep learning algorithms are also able to distinguish between different types of bifurcations as they are being approached which can offer vital information regarding the potential for catastrophic collapse in CHES.

**3.2 Social data for early warning signals**

In CHES models, the strength of EWS from environmental data has been shown to be muted compared to EWS from environmental systems not coupled to a human system (Bauch et al., 2016) or the same system with weak coupling between the human and environmental subsystems (Richter and Dakos, 2015). This is likely due to the

effects of human behavior acting to mitigate variability in the environmental system, for example, rarity-motivated valuation creates a negative feedback loop where incentives to mitigate increase as the environment becomes further depleted, serving as a mechanism to avoid collapse. The muting of EWS provides a unique challenge for monitoring tipping events in CHES using environmental data, especially as they occur more frequently in these coupled systems as discussed in Section 2. There are a small number of studies that have directly compared the strength and efficacy of EWS between various state or auxiliary variables in CHES models. In these studies, generic EWS from data in the human system were shown to be the only reliable indicators of the coupled system approaching a tipping point. Examples of human data used include the fraction of conservationists in a forest cover model (Bauch et al., 2016), average profits by resource harvesters, and catch per unit effort common-pool resource models (Lade et al., 2013; Richter and Dakos, 2015). In agreement with generic methods, a state-of-the-art machine learning algorithm for EWS showed higher success in detecting tipping events generated from a coupled epidemiological model using pro-vaccine opinion in the human system compared to total infectious in the epidemiological system (Bury et al., 2021). It is possible that the state variable most sensitive to the forcing parameter may exhibit the strongest EWS, as seen in experimental work on tipping points in a lake food web. In this system, data from the species that had a direct trophic linkage to a driver of the tipping event (predators added to the food web) exhibited EWS earlier than those that were farther removed from the driver (Carpenter et al., 2014). If this is the case, human drivers of tipping points would most directly affect the human system, and EWS should still be stronger using social data.

The improved reliability of EWS from social data demonstrated through CHES models shows a significant promise for monitoring resilience in CHES through the analysis of socio-economic data (Box 1.4). This confers a practical advantage as socio-economic data availability is growing faster than ecological data (and perhaps even environmental data despite the growth of publicly available satellite data) on account of the era of digital social data (Ghermandi and Sinclair, 2019; Hicks et al., 2016; Lopez et al., 2019; Salathé et al., 2012). Some examples of this are monitoring profits tied to resource extraction as well as using sentiment analysis on social media data, such as the number of tweets in a given area raising concern over the health of a coupled environmental system. Furthermore, citizen science not only generates environmental data but also provides social metadata through the participation of users who monitor specific areas. Leveraging existing platforms like CitSci.org, we can use this data to estimate trends in conservationist frequency over time (Wang et al., 2015). This approach allows for the implementation of real-time monitoring of environmental systems using data that is currently being generated, reducing the need for extensive knowledge or complex mechanistic models of the system. With the potential social data offers for use with EWS, it is important to note that much of the traditional social data, often conducted through national or regional surveys, do not provide fine-grained spatial or temporal resolution. On the other hand, novel methods that use social media data can solve the resolution issue, but may not accurately represent the population it is being used to model (Hargittai, 2020). These challenges may be addressed through a compound approach that uses hybrid time series generated from multiple types and sources of social data (Rosales Sánchez et al., 2017).

## 4 Conclusion and future directions

### 4.1 Summary of main points

From a wide range of examined theoretical models, we are able to gain insight into human drivers that lead to tipping events in CHES systems. Many social interventions, such as reducing mitigation costs and extractive effort, or increasing the time horizon in decision-making, lead to beneficial tipping events, regardless of the system modeled. The beneficial effect of these interventions is intuitive; however, non-linear responses manifested as tipping events may not be as evident. Mitigation costs can be reduced through subsidies for land preservation and green technology, and extraction effort through limits on land development and the expansion of protected natural areas (i.e. the Haudenosaunee-led protection of the Haldimand Tract) (Forester, 2021), and by increasing time horizons through passing long-term legislation that centers the well-being of human and environmental systems such as the Green New Deal (Galvin and Healy, 2020). These policy interventions become more difficult to implement at large scales, and models that are tailored to global coordination problems can give us insight into how institutions can work together to rapidly mitigate looming threats, such as the current climate crises we are facing (Karatayev et al., 2021).

Other human behaviors and social processes are much more nuanced and system-specific in how they affect tipping points. For example, models show that rarity-motivated valuation can act to detrimentally tip the environmental system into a depleted state when it crosses both a lower and (counterintuitively) an upper threshold value. This was illustrated most clearly in the example of forest cover in the paper by Bauch et al. (2016). Social norms, especially when majority-enforcing, increase the likelihood of tipping points through the emergence of bistable regimes that are made up of both sustainable and unsustainable environmental equilibria. The extent of coupling between the human and environmental system as well as the speed of social change relative to environmental change can have different effects depending on whether the model is human-extraction or human-emission. Interventions related to human valuation and social norms are much more difficult to implement as they require a deeper mechanistic understanding of how to influence social dynamics and may also have ethical considerations.

The models we reviewed also show that greater structural complexity via the number and diversity of human traits as well as the number of social connections can increase the potential for tipping points and mask social dynamics making these transitions much harder to predict. The modeling literature has only explored a small sliver of the space of possible choices regarding assumed social structure and the types of environmental models coupled to them. For example, the vast majority of models only allow for a binary choice in human behavior and adaptive social networks have only recently been incorporated, with limited mechanisms of re-wiring and types of coupled environmental systems. Consequently, we still have much to learn on how shifting underlying social structures acts as a driver of tipping events. This is especially true in human-emission models which are important to improving our

understanding of how our social structures affect pressing global issues such as pollution and climate change. Even if we include more diverse and realistic social structures and processes, CHES are composed of many non-linear feedbacks and contain high levels of uncertainty, and the reality is that we may not be able to have a complete mechanistic representation through models. EWS from empirical data show great potential in predicting tipping events without requiring a full understanding of the system being monitored. There have been many advances in using state-of-the-art machine learning algorithms to provide accurate EWS from 1-D time series (Bury et al., 2021; Deb et al., 2022), and very recent work is now developing similar techniques to predict tipping events from spatial data (Dylewsky et al., 2022). As synthetic data from models have shown the value of EWS from social data, it is likely that applying these techniques to diverse and hybrid empirical social datasets can vastly improve our ability to predict tipping events caused by human drivers in the future.

## 4.2 Future work in CHES modeling

There are many social phenomena that are not commonly included in CHES models, yet may be important in furthering our understanding of tipping points within these systems. We know that inequality in human systems plays a large role in individuals' risk perception and ability to engage in pro-environmental behavior (Gibson-Wood and Wakefield, 2013; Pearson et al., 2017; Quimby and Angelique, 2011; Rajapaksa et al., 2018) and have mentioned two CHES models that incorporate wealth inequality in a human-climate system (Menard et al., 2021; Vasconcelos et al., 2014). However, more studies explicitly investigating the role of inequality could offer some valuable insight into interventions that can be more effective in benefiting both the environment and the most vulnerable in human systems. This could be complemented by incorporating social biases where perceptions of risk are linked to an individual's socio-economic status, and detrimental environmental outcomes are experienced disproportionally by vulnerable communities as is commonly observed globally (Banzhaf et al., 2019; Boyce, 2007). Future models could allow for alternatives to the common modeling assumption where individuals act in their own self-interest, for example by incorporating other-regarding preferences into utility functions so that individuals value their neighbors' well-being along with their own (Dimick et al., 2018). These models could also look at grassroots redistribution of wealth allowing us to explore the effects of alternative social value systems on the environment (Tilman et al., 2018).

Stochasticity (noise), especially regarding drivers of tipping points can significantly affect system dynamics including when tipping points occur. Although many CHES models are deterministic, recent work has shown that increasing noise can lead to earlier tipping (Willcock et al., 2023), or in other cases, increase the duration of time the environmental system can persist before becoming extinct (Jnawali et al., 2022). These contradictory results warrant further work in understanding how different types of noise and their magnitude within drivers of tipping events affect the resilience of these systems. With stochasticity comes uncertainty, and in real-world systems, it is impossible to know with precision the extent of social change required to bring about a beneficial or avoid a

detrimental tipping point. This uncertainty around our knowledge of system thresholds adds an additional challenge in both agreeing upon and following through with policy that promotes sustainable futures while taking into account potential tipping points. Experimental games have shown that high threshold uncertainty can promote the collapse of a shared resource, often through an increase in free-riding behavior (Barrett and Dannenberg, 2014, 2012). On the other hand, field experiments in fishing communities have shown that high uncertainty can promote cooperation and sustainable resource use (Finkbeiner et al., 2018; Rocha et al., 2020). Theoretical models show that increased uncertainty can lead to increased mitigative behavior if the shared resource is highly valued; however, for low-valued resources, increased uncertainty can deter mitigation, putting the persistence of the shared resource at risk (Jager et al., 2000; McBride, 2006). Uncertainty around thresholds is unavoidable, further motivating the need to offer additional incentives for mitigative action on institutional scales, rather than solely the threat of environmental collapse. In systems where uncertainty can promote mitigative action, increased communication and awareness campaigns around this threshold uncertainty could be useful to incorporate into policy.

This review has focused primarily on the effects of single drivers; however, research on multiple co-occurring human drivers of tipping events, while more analytically challenging, could offer a holistic understanding of how these drivers interact. A recent study has shown that multiple drivers can both reduce the time until tipping or lead to a tipping point that would not occur with a single driver (Willcock et al., 2023) and there is already a large body of empirical work exploring the diversity of these drivers which can be used to inform future CHES models (Jaureguiberry et al., 2022; Maciejewski et al., 2019; Millennium Ecosystem Assessment, 2005). Finally, as the majority of the studies in modeling tipping points have focused on slow gradual changes in the driver, fast changes require further research as they can exhibit very different tipping behavior (Ashwin et al., 2012). CHES models ubiquitously exemplify the phenomenon of tipping points, which often occur through drivers in the human system. Although these models offer valuable insight in understanding key feedbacks and qualitative behavior, their predictive power is limited. Additionally, as many model findings can depend on the type of system modeled as well as assumptions in the model formulation, translating this work into policy remains a significant challenge. However, further work in both diversifying model systems and assumptions paired with research in universal real-time indicators of EWS shows considerable promise in both improving our understanding and predicting human drivers of tipping events in the environment.

**Author contribution.** I.F.: visualization, writing—original draft, writing—review and editing; C.T.B.: visualization, writing—original draft, writing—review and editing; M.A.: conceptualization, funding acquisition, supervision, visualization, writing—original draft, writing—review and editing.

**Competing interests.** The authors declare that they have no conflict of interest

**Funding.** This research was supported by the Natural Sciences and Engineering Council of Canada (Discovery grants to both M.A and C.T.B), the Canada First Research Excellence Fund (to M.A.), and in part by the International Centre for Theoretical Sciences (ICTS) for the program "Tipping Points in Complex Systems " (code: ICTS/tipc2022/9) in which M.A. and C.T.B. participated.

 **Appendix**

| Authors | Year | Title | System of study |
|---|---|---|---|
| Sethi & Somanathan | 1996 | The evolution of social norms in common property resource use | Common pool resource |
| Satake et al. | 2007 | Coupled ecological–social dynamics in a forested landscape: Spatial interactions and information flow | Land use |
| Iwasa et al. | 2007 | Nonlinear behavior of the socio-economic dynamics for lake eutrophication control | Lake eutrophication |
| Suzuki & Iwasa | 2009 | The coupled dynamics of human socio-economic choice and lake water system: the interaction of two sources of nonlinearity | Lake eutrophication |
| Iwasa et al. | 2010 | Paradox of nutrient removal in coupled socioeconomic and ecological dynamics for lake water pollution | Lake eutrophication |
| Figueiredo & Pereira | 2011 | Regime shifts in a socio-ecological model of farmland abandonment | Land use |
| Tavoni et al. | 2012 | The survival of the conformist: Social pressure and renewable resource management | Common pool resource |
| Lade et al. | 2013 | Regime shifts in a social-ecological system | Common pool resource |
| Iwasa & Lee | 2013 | Graduated punishment is efficient in resource management if people are heterogeneous | Fishery |
| Richter et al. | 2013 | Contagious cooperation, temptation, and ecosystem collapse | Common pool resource |
| Richter & Grasman | 2013 | The transmission of sustainable harvesting norms when agents are conditionally cooperative | Common pool resource |
| Barlow et al. | 2014 | Modelling interactions between forest pest invasions and human decisions regarding firewood transport restrictions | Pest |
| Vasconcelos et al. | 2014 | Climate policies under wealth inequality | Climate |
| Ali et al. | 2015 | Coupled human-environment dynamics of forest pest spread and control in a multipatch, stochastic setting | Pest |
| Sugiarto et al. | 2015 | Socioecological regime shifts in the setting of complex social interactions | Common pool resource |
| Wiedermann et al. | 2015 | Macroscopic description of complex adaptive networks coevolving with dynamic node states | Private resource |
| Richter & Dakos | 2015 | Profit fluctuations signal eroding resilience of natural resources | Common pool resource |
| Schlüter et al. | 2016 | Robustness of norm-driven cooperation in the commons | Common pool |

| | | | resource |
|---|---|---|---|
| Weitz et al. | 2016 | An oscillating tragedy of the commons in replicator dynamics with game-environment feedback | Common pool resource |
| Bauch et al. | 2016 | Early warning signals of regime shifts in coupled human–environment systems | Forest |
| Henderson et al. | 2016 | Alternative stable states and the sustainability of forests, grasslands, and agriculture | Land use |
| Sugiarto et al. | 2017 | Social cooperation and disharmony in communities mediated through common pool resource exploitation | Common pool resource |
| Barfuss et al. | 2017 | Sustainable use of renewable resources in a stylized social–ecological network model under heterogeneous resource distribution | Private resource |
| Lafuite et al. | 2017 | Delayed behavioral shifts undermine the sustainability of social–ecological systems | Land use |
| Lindkvist et al. | 2017 | Strategies for sustainable management of renewable resources during environmental change | Common pool resource |
| Osten et al. | 2017 | Sustainability is possible despite greed - Exploring the nexus between profitability and sustainability in common pool resource systems | Common pool resource |
| Sigdel et al. | 2017 | Competition between injunctive social norms and conservation priorities gives rise to complex dynamics in a model of forest growth and opinion dynamics | Forest |
| Sugiarto et al. | 2017 | Emergence of cooperation in a coupled socioecological system through a direct or an indirect social control mechanism | Common pool resource |
| Thampi et al. | 2018 | Socio-ecological dynamics of Caribbean coral reef ecosystems and conservation opinion propagation | Coral reef |
| Chen & Szolnoki | 2018 | Punishment and inspection for governing the commons in a feedback-evolving game | Common pool resource |
| Drechsler & Surun | 2018 | Land-use and species tipping points in a coupled ecological-economic model | Land use |
| Geier et al. | 2019 | The physics of governance networks: critical transitions in contagion dynamics on multilayer adaptive networks with application to the sustainable use of renewable resources | Private resource |
| Hauert et al. | 2019 | Asymmetric evolutionary games with environmental feedback | Common pool resource |
| Lin & Weitz | 2019 | Spatial interactions and oscillatory tragedies of the commons | Common pool resource |
| Sigdel et al. | 2019 | Convergence of socio-ecological dynamics in disparate ecological systems under strong coupling to human social systems | Common pool resource |

| | | | |
|---|---|---|---|
| Bury et al. | 2019 | Charting pathways to climate change mitigation in acoupled socio-climate model | Climate |
| Shao et al. | 2019 | Evolutionary dynamics of group cooperation with asymmetrical environmental feedback | Common pool resource |
| Barfuss et al. | 2020 | Caring for the future can turn tragedy into comedy for long-term collective action under risk of collapse | Common pool resource |
| Tilman et al. | 2020 | Evolutionary games with environmental feedbacks | Common pool resource |
| Muneepeerakul & Anderies | 2020 | The emergence and resilience of self-organized governance in coupled infrastructure systems | Water use |
| Sun & Hilker | 2020 | Analyzing the mutual feedbacks between lake pollution and human behavior in a mathematical social-ecological model | Lake eutrophication |
| Mathias et al. | 2020 | Exploring non-linear transition pathways in social-ecological systems | Common pool resource |
| Phillips et. al | 2020 | Spatial early warning signals of social and epidemiological tipping points in a coupled behavior-disease network | Epidemic |
| Menard et al. | 2021 | When conflicts get heated, so does the planet: coupled social-climate dynamics under inequality | Climate |
| Phillips & Bauch | 2021 | Network structural metrics as early warning signals of widespread vaccine refusal in social-epidemiological networks | Epidemic |
| Holstein et al. | 2021 | Optimization of coupling and global collapse in diffusively coupled socio-ecological resource exploitation networks | Private resource |
| Farahbakhsh et al. | 2021 | Best response dynamics improve sustainability and equity outcomes in common-pool resources problems, compared to imitation dynamics | Common pool resource |
| Yan et al. | 2021 | Cooperator driven oscillation in a time-delayed feedback-evolving game | Common pool resource |
| Müller et al. | 2021 | Anticipation-induced social tipping: can the environment be stabilised by social dynamics? | Climate |
| Milne et al. | 2021 | Local overfishing patterns have regional effects on health of coral, and economic transitions can promote its recovery | Coral reef |
| Moore et al. | 2022 | Determinants of emissions pathways in the coupled climate–social system | Climate |
| Bengochea Paz et al. | 2022 | Habitat percolation transition undermines sustainability in socialecological agricultural systems | Land use |

824

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
