# Peer review of "Tipping points in coupled human-environment system models: a review"

_EGUsphere, 2023_

## Author Response (AR1)

**Public justification (visible to the public if the article is accepted and published):**
*The three reviewers and I agree that this paper has potential to be an important contribution to the literature. However the reviewers are consistently clear that many aspects of the manuscript need to improved, including*
*- clarifying the scope of the review, such as whether it is limited to models and certain types of human behaviour. This problem is most acute in the title, but also needs clarification elsewhere.*
*- more clearly stating the insights gained from the review -- throughout abstract, results and discussion. A good review should be more than a collection of examples but also produce some 'results' or novel insights.*
*- including some brief information about the review method. If making claims about a body of literature, how can the reader be confident you have systematically sampled the literature? Methods could range from a systematic literature review to snowball sampling*

**I also suggest**
*- Clarifying early on what exactly you mean by 'drivers', since it is such a key term in the article. In the DPSIR framework and in systems thinking, it usually means some external forcing is affecting the system. However your discussion point #3 in AC2 appears to take "structural complexity" as a driver. (If I have misinterpreted and you do not consider it a driver, I wonder whether it's appropriate that only one of your four discussion points deals with drivers?)*

Thank you for these comments. We have clarified the scope by changing the title and adding text to section 1 and the beginning of section 2.

We have made significant changes to the abstract, introduction and discussion, as well as other changes throughout the rest of the manuscript to better highlight insights gained throughout this review.

We have added a description of our snowball review method at the end of section 1, with mention in the abstract as well.

Our definition of 'drivers' has been clarified at the end of section 1.

**RC1**

*Title. I think the title should make clear that this is a review of drivers of tipping points in coupled human-environment systems MODELS – as opposed to drivers inferred from empirical data of these systems. This is an important distinction to make.*

We will change the title to "Drivers of tipping points in coupled human-environment systems models: a review"

*Definition and examples of tipping points. For me, tipping points or regime shifts are fascinating because these abrupt, almost irreversible changes are induced by gradual changes in some components of the systems. Some examples presented in the Introduction, however, seemed to focus on those cases in which abrupt, big changes (big changes in parameters) lead to big changes in system outcomes, which are to be expected and do not capture the fascinating aspect of tipping points. Personally, I see a tipping point as closely related to bifurcation: the coupled system must go through changes in the number and/or nature (stability) of its stable equilibria. In cases where noise is considered, a tipping point may be said to happen when the system cross into a different basin of attraction associated with an alternative stable state. However, some of the examples in this review seem to simply be cases where changes happen quickly – which I don't think is sufficient to be classified as a tipping point. Does your definition of a tipping point require bifurcation and/or crossing a boundary of a basin of attraction or not? I think it should – even must - but this was not clear. Please clarify.*

We mentioned "fast changes" and "disturbances" only in one sentence in the Introduction while presenting another authors' review paper. We take this as what this reviewer is picking up on as "big changes" but we in fact never use that term. Regardless, these kinds of changes are not referenced at all in our manuscript afterwards, and to avoid further confusion, we will remove the following text from that sentence: "Gradual  changes in system parameters, for example, the rate of resource extraction,  can cause the system to abruptly transition between these states." In terms of the question: "Does your definition of a tipping point require bifurcation and/or crossing a boundary of a basin of attraction or not?", the answer is a definitive "yes", however we see now why there there could be confusion about this because some of the qualitative examples we gave in the Introduction have not (yet) exhibited bifurcation and/or crossing a boundary of a basin of attraction (i.e., the rebound of the bald eagle and wolf populations following the enactment of conservation laws, as well as the banning of DDT regarding the declining eagle populations). We will remove examples that have not been fully studied.

*Better connections among reported findings are need. Too many paragraphs have the following pattern: These papers say this on the topic, while those papers say that, and the reader is left hanging and confused at the end of these paragraphs. I feel that a useful review should offer more than a list of findings.*

We will restructure the sections which appear to have a repeated pattern and provide more commentary on past studies. We will also restructure the paper to highlight our main findings.

*And when a generalized statement was attempted, I don't think it was done with enough care. For example, Section 2.5 Rates of social change and time horizons – I agree with the authors that this is a critical component of a CHES model, but I feel that it was presented too simplistically.  On lines 172-173, for example, the authors stated "…decreasing the relative speed of social dynamics brings about positive tipping point…" – is that true always?  What if the initial extraction rate is too high?  The low speed of social dynamics can translate to insufficient adaptation in the behavior, which would eventually lead to a system collapse.*

This is a good point regarding low rates of learning and environmental collapse (tipping in the environmental system). We will qualify this statement to say what is only true of the papers which we found/surveyed in our review.

*The effects of stochasticity or noise is not sufficiently addressed.  There was some discussion of the effects of noise in Section 4 (about early warning signals) but hardly any in the previous sections. So most of the models discussed in Sections 2 and 3 were deterministic models only? There is a large literature on noise-induced transitions, which are closely related to tipping points, but this was not sufficiently addressed.*

We agree that noise is an important driver, but there are not many papers looking at this in human-environment models. We therefore find it out of the scope of this review, and see this as an opportunity to point this out as a gap in our 'future work' section. Thank you.

All technical corrections will be made in the text

**RC2**

*The title is misleading - The title should make it clear that this is a review article.*

We will change the title to "Drivers of tipping points in coupled human-environment systems models: a review"

*There are many broad statements that require further support with references, e.g. "There are many historical cases of human-induced tipping points that have drastically affected the trajectories of coupled environment systems and these effects can be both beneficial as well as catastrophic." Such a broad, strong statement requires many supporting references and qualification.*

We will add citations to that sentence as well as other broad statements made throughout the text.

*The discussion is quite general and descriptive and lacks a key unifying argument. What are we supposed to learn? What is the take away message beyond a loosely structured catalog of examples?*

*Overall, the paper lives somewhere between discussing specific models and discussing general principles and doesn't connect the two very well. I don't know if readers of this journal will get much out of it - I am guessing they will already be aware of most or all of the material covered. All of the points made in the "Conclusion and future directions" section are very well known and suggestions for future work are not very inspiring.  I think the authors need to give some critical thought to what the key message is.*

In terms of "What are we supposed to learn?": 1. There are counterintuitive tipping points in human-environment systems. 2. Many tipping points are system-specific in terms of drivers, especially in regards to input vs output-limited models. 3. Higher structural complexity in the social system increases the likelihood of tipping points. 4. Social time series data offer great potential for early warning signals, especially when using state-of-the art techniques. These four may be useful in terms of interventions by society/policy-makers. We will restructure and rewrite our Discussion section to highlight these and other specific findings.

*1) The examples on human behavior aren't about human behavior. "Coupling strength" is a system feature. What is the behavior?  "Rarity motivated conservation" is an outcome.  What is the human behavior that generates it?*

In terms of "what is the behaviour?", it is often the extraction or pollution rate by humans. We made some changes in wording to clarify

The human behaviour that generates what we called "Rarity motivated conservation" represents the extent to which humans change their [extractive] behaviour in response to the environment reaching a depleted state because of changing valuation. We will change this wording. Thank you.

*2) Figure 1 has appeared countless times with different labels. This is a general feature of frequency-dependent selection.  There is nothing particular about human behavior here other than the state variable in some sort of replicator equation is labeled "opinion".*

The mechanism behind Figure 1 is common but the applications to CHES is under-appreciated in our opinion.  Social norms are sometimes treated as only capable of supporting conservation in some cases.  In other research it is assumed they can only suppress conservation.  We wish to highlight human-environment systems–often at larger spatial and temporal scales–in which social norms can either support or suppress mitigation depending on the initial majority behaviour.  This generates classic bistability patterns as we show in FIgure 1. We will better highlight this finding in the revised text and Figure 1.

*3) Sections 2.3-2.5 are likewise not about human behavior but, rather, are about social dynamics. Maybe change the title of section 2 to "Social processes that may generate tipping points" or something like that.*

We will change the section title to "Social processes that lead to tipping points in CHES models"

*4) Section 3 discusses results specific to a particular model (Figure 2). This is difficult to follow - the actual model needs to be presented. The same is true of Figure 3.*

We will extend the explanation of the model for Fig. 2. We will not present the actual model because it would take a few paragraphs to fully explain and is not needed to get the point we want to make with the figure which is how the rewiring probability has a tipping point at low and high levels. For Fig. 3, the models are unspecified ecological models, and their specifics are less relevant since the figures focus is on comparing deep learning EWS to traditional methods. The figure caption has been changed to include additional clarification.

**RC3**

*The authors present an overview of typical parameters in coupled human-environment system models that can cause a tipping point. In my view, this article has the potential to make a nice contribution to the field of coupled systems modeling, viewed from the lens of social tipping points. However, some issues must be addressed beforehand.*

*First, the authors subsume a specific kind or style of coupled human-environment system models as presented in the introduction. This should be stated more transparently. Not all coupled systems models use or should use a form of social learning. This is important to contextualize since the factors the authors then eventually discuss heavily depend on this choice of coupled systems modeling style. The end of the introduction's second-to-last paragraph, beginning with "Common factors in the utility function are the rate of social learning," should additionally be supported with references. The authors should also describe briefly how they came about their review and choice of aspects.*

We will add a statement acknowledging CHES models without social learning exist and we have chose to focus on social learning models. We will also add a sentence mentioning that we chose common social processes in CHES models at the end of the first paragraph of section 2. After our statement on the why for this paper i.e. "...we aim to deepen our understanding of human-induced tipping points through CHES models…", we also include how we conducted our literature review.

*Second, and a possible consequence of the first point, I doubt that the aspects the authors discuss comprise a complete list. For example, the role of inequality, biased perceptions, uncertainty, risk levels, and other-regarding preferences can be significant in some situations but are not discussed as individual points, not even in the outlook for future work. In some cases, there might be few works; in others, a different style of modeling coupled systems*

*already covers these factors. Thus, I invite the authors to contemplate which factors are generally relevant and should be studied (in possible future model studies).*

We do mention two models with wealth inequality, under section 3.2 and agree that the list of points is not exhaustive and ones the reviewer lists are interesting to discuss. We will take up the reviewer's invitation "to contemplate which factors are generally relevant and should be studied (in possible future model studies)."

*Third, a critical reflection on the robustness of the collected results is missing. For example, I wonder to what extent the occurrence of tipping points in these kinds of models might be a result of the fact that the strategy space is parameterized between two possible opinions/actions/moves: behaving sustainably vs. non-sustainably and the fact that social learning process assumes that agents share the same preferences. Thus, when the conditions change such that sustainable behavior becomes advantageous, the whole population changes. Likewise, is the observation that high structural complexity increases the potential for tipping points a result of the increased dimensionality of the system?*

You are right, in many cases, the answer to what drives tipping points is "it depends". For example, regarding "*is the observation that high structural complexity increases the potential for tipping points a result of the increased dimensionality of the system?*" the answer is: it is not known. However, we did find some commonalities in other cases, where certain processes tend to drive certain types of tips in the range of studies we included in the review. We can highlight this in a more critical reflection in the Discussion and better delineate the cases where commonalities are found to the "it depends" cases.

*Fourth, the structure of the review does not become clear to me. Section 2, talking about ASPECTS of human behavior, is so general that the aspects from Section 3 could go in there, too. The division is confusing since social traits, which I would better call the strategy space of the model, are nothing that can only be varied in agent-based models. The replicator dynamics are not limited to a two-strategy setting. And the role of Section 4 on early warning signals needs to be clarified. In model studies, we typically vary the parameters and observe the tipping point directly. If the point is to highlight empirical approaches, this should be addressed upfront. Overall, the structure of the manuscript should be clarified and introduced better.*

We will change the wording of "aspects of human behaviour" to "social processes". Section 3 looks at structure via strategy space and spatial structure - we could make that more explicit in the text. For tipping points, we will mention that they are used on both empirical and synthetic data. Overall, we will clarify the structure of the manuscript and introduce it better as this reviewer suggests.

*Fifth and last, after reading this manuscript, I have the impression that all parameters in coupled system models are or at least can be a tipping element. The review will gain in value if it contrasts such abrupt transitions with processes of gradual change in coupled human-environment system models.*

Like any sufficiently high dimensional dynamical system, we agree that in principle, any parameter combination evolving in parameter space has the potential to cause a tip in these systems. However, in some cases, the empirical constraints on the parameter space preclude that. We will better convey this in the revision, and also contrast tipping events with gradual changes as per the reviewer's suggestion, with specific reference to rebounding natural populations caused by resilience (negative feedback) in example systems

---

## Author Response (AR2)

**Public justification (visible to the public if the article is accepted and published):**
Reviewers 1 and 3 indicate that while the manuscript is developing on the right track, substantial refinement is still required. Reviewer 2 is happy with the manuscript in its current state.

Reviewer 1 highlights that the paper's key messages could be expressed more clearly. They also question whether the described literature review method actually matches what was done. I would add to their comments that
- there is some inconsistency between the abstract, where only snowball sampling is mentioned, and the main text where both snowball sampling and search terms are mentioned.
- the search terms are very broad and do not mention models or modelling
- a review would conventionally include an appendix with a list of the papers found and how these papers were coded or analysed

Reviewer 3 raises more general questions about whether the field is sufficiently mature for a review and calls for more nuanced writing around some points.

Thank you for these comments. We made a table with a list of all the papers reviewed; however, we do not feel as though it is adding anything significant since it is a subset of our reference list. None of the papers were coded or analyzed. We added our use of search terms in the review process and further explained our review methodology. We have addressed reviewer 3's concerns in the text (see response to reviewer 3 for more details).

**Reviewer #1**

The manuscript's structure is now more apparent, although a brief explanation at the end of the introduction would still be helpful.

We have added a sentence at the end of the introduction laying out the structure of the text (line 164)

A significant point of obscurity after the revision is the authors' review methods. The authors write they did a Google Scholar search with the following search key:

> ('human environment system' OR 'socio-ecological system' OR 'social ecological system' OR 'human ecological system' OR 'human natural system') AND ('tipping' OR 'regime shift' OR 'bifurcation')

I did a quick Google Scholar search myself using the tool "Publish or Perish." Counting only articles with at least one citation, I obtained more than 750 results. What did the authors do with that many results? It also seems logically inconsistent to write later that the authors focus primarily on models with social learning processes.

From the initial results, we filtered papers that were dynamical models showing clear signs of tipping behaviour. This has been clarified in the text (line 160)

Furthermore, from the authors' response, I understand that the four key contributions of their work are

1. There are counterintuitive tipping points in human-environment systems.
2. Many tipping points are system-specific in terms of drivers, especially regarding input vs output-limited models.
3. Higher structural complexity in the social system increases the likelihood of tipping points.
4. Social time series data offer great potential for early warning signals, especially when using state-of-the-art techniques.

First, I encourage the author to revise also the paper to that level of clarity.

We have included a box with images to further highlight these key findings with references to them in the parts of the text where these points are discussed.

Regarding point (1), I do not have any intuition regarding rarity-motivated valuation. Thus, I don't find a lower threshold value leading to depletion counterintuitive. And it is pretty intuitive to me that social norms increase the likelihood of a tipping point. I would be interested in which counterintuitive tipping points the authors have considered and why they are counterintuitive.

The high threshold is the one that is counterintuitive for rarity-motivated valuation and thank you for pointing out the error. This is counterintuitive because the social system will be more sensitive when responding to a resource reaching low levels (i.e. by reducing harvesting rates), which intuitively should lead to the persistence of the resource and not increase the likelihood of collapse, which is in fact the case in many models. We have clarified this in the text (line 234)

We are assuming that social norms increasing the likelihood of tipping points isn't immediately intuitive to the reader who is less familiar with social norms, alternative stable states, and hysteresis.

Other human drivers of tipping points that aren't immediately intuitive are the coupling strength as well as the relative speed of social and environmental dynamics. Perhaps "counterintuitive" is not the best word in all cases, as many of these drivers of tipping events are not obvious or the reader may not have an intuition, and the wording in the text reflects that.

Regarding point (2), it is to be expected that many tipping points are system-specific. I would be interested in the authors' thoughts on why this insight is one of the manuscript's key contributions to the literature.

This review points out ways in which these tipping points are system-specific. As it is a review, it is not contributing any new information, but summarizing common findings, of which this is one. One contribution that comes from our synthesis which is not prevalent in the literature is showing how some of these specificities can be generalized to input and output-limited (or

human-extraction vs human-emission) models, especially regarding relative rates of social and environmental change, as well as coupling strength.

Regarding point (3), which I find noteworthy, I would be interested in why this phenomenon occurs. Do the authors summarize what they observed in their review, or could more be said about the underlying reason? Higher structural complexity could also lead to higher smoothing and less tipping.

We summarize our observations in this review; however, we also note that since there are not many CHES models that explicitly look at the effects of structural complexity, it is difficult to make any generalizable statements about the underlying reason for this observation (also, one of the other reviewers cautioned us against making too many generalizations).

(No comments regarding point (4))

Last, the track-changes file is weirdly formatted. For example, Section "2.1 Coupling strength" is deleted, then appears again as a novel inclusion after one paragraph. It makes the implemented changes look substantial when there is a lot of red in the tracked changes file. However, with such practices, the value of a track-changes file diminishes.

**Reviewer #3**

In this revision, the authors have clarified the scope and focus of the review, including changing the title and made some improvements throughout the piece. However, the clarified scope appeared to be quite narrow and restricted for a review. Most of the models reviewed assumed that dichotomous human behavior (i.e., choosing between two strategies) and were built on replicator dynamics (social learning), making the scope of the review rather limited. On models with more strategies, the authors wrote "there are relatively few models that explicitly compare the complexity of social traits and their effect on tipping points" (lines 349). Similarly, on the role of stochasticity as a driver of tipping points, the authors responded in Response to Review that "there are not many papers looking at this in human-environmental models." These comments led me to wonder if a review such as this was premature – the field is still growing and the models that address important aspects of tipping points in coupled human-environmental systems are yet to be developed. Perhaps, this was why the scope of this review feels narrow.

We reviewed 52 papers that included CHES models with tipping points. We feel this is sufficient. In comparison published reviews (Filatova *et al.*, 2016) reviewed 16 models, (Schlüter *et al.*, 2012) reviewed ~65 models, however, this was an exhaustive review of CHES models, many of which did not exhibit tipping behaviour, (Tallis & Kareiva, 2006) have 25 citations in total, (Wange *et al.*, 2018) have 34 citations in total. How many models cited does the reviewer think is necessary?

The reviewer chooses to highlight the two areas where we explicitly highlight where papers are lacking in the literature (for the purposes of identifying gaps in the literature, a vital component of any comprehensive review) to make a larger claim that there is not enough literature at large, but in fact, we go through quite a lot of areas for which there are many studies. One of the points of reviews is to highlight knowledge gaps and that is what we do with respect to the diversity of models and inclusion of social heterogeneity, for example.

Specific comments

Abstract: "…gradual changes to the human system" – Why only the human system? What about gradual changes to the environmental system?

This paper specifically addresses human drivers of tipping points in coupled human-environment systems. We make this clearer in the abstract (line 9).

Definition and examples of tipping points. Let me reiterate my position on tipping points. For me, tipping points or regime shifts are fascinating because these abrupt, almost irreversible changes are induced by gradual changes in some components of the systems. Reading through the revised manuscript, I found myself still asking the question "what is the small gradual change here?" in a few examples provided. For example, what is the small gradual change in the case where British colonial government imposed their own rules in Northern India? A government imposing new rules does not feel gradual to me. In the ozone example, the authors wrote "Then when policy was passed, industry shifted abruptly to producing CFC alternatives, which led to a tipping point": Again, a policy being passed does not feel gradual to me.

How valid/generalizable are some of the authors' statements? There is a larger challenge of this piece. Many of the reported findings are model-specific – the authors said as much at a few places. It is difficult to judge validity and generality of some of the statements made without seeing the model details. Sometimes, general statements were made with citing only one or two models.

We have broadened our definition of tipping events to include tipping points that have been crossed through larger perturbations. This is in line with tipping point literature (a phenomenon known as shock-tipping or s-tipping) (Halekotte & Feudel, 2020; Boettiger & Batt, 2020) and describes many historical examples of social tipping as the introduction of government policy often plays a role in CHES tipping events. Much of the literature on social tipping references the introduction of policy as a key driver in historical examples (Lenton *et al.*, 2022). We have clarified this in the text (line 85) and included a new figure to illustrate this (Fig. 1).

We synthesized findings across papers in response to the comments from Reviewer 1 who asked for more than a list of specific findings. Where we make a claim to generality, the claim is as general as the cited papers support. We have ensured that we cite all the relevant papers, where such claims are made.

Fig 1: Please provide a reference of the model on which the diagrams between emissions and proportion of mitigators were based so that the interested reader can check out the model's details.

This is our own conceptual model used to illustrate the effects of CHES feedbacks, specifically in relation to social norms.

Fig 2: According to the caption, panel b corresponded to an ODE model. I think many readers would be confused: How can there be "rewiring" in a non-network ODE model? If I'm not mistaken, the ODE model in panel b was an approximation the true network model in panel a. In the context of this review on tipping points, I can understand the inclusion of panel a (the importance of rewiring), but what is the contribution of panel b to this review? It seemed to me that the point of panel b in the original paper (Wiedermann et al., 2015) was to showcase how to mathematically derive a macroscopic approximation of the true network model. It was more mathematical in nature, rather than substantial for the understanding of tipping points. There were only 3 figures in this review – it was unclear why panel b was given priority for the limited space.

We have removed panel b for clarity.

"Input-limited" and "output-limited" models. The terms "input-limited" and "output-limited" were first introduced in Section 2.1, line 175. Are these widely used terms in the field? I am not familiar with them. If so, please provide references when they were first introduced here. If these were something that the authors proposed, I must say that the terms were not very intuitive/accurate for me.

Did "coupling" refer to anything other than "extracting" in "input-limited models" and "polluting" in "output-limited models" in this review? If so, please provide examples. If not, please clarify that (using a general term to describe specific things can be confusing).

We found these terms in the literature but we agree that these terms may not be widely used. We have decided to instead use the terms 'human-extraction' vs. 'human-emission'.

Lines 292-293, the authors wrote "The reduced speed of social change leads to beneficial outcomes as the resource is allowed more time to stabilise as decisions regarding extractive levels occur." – is that true always? What if the initial extraction rate is too high? In fact, I objected to this oversimplified statement in my previous round of review. In their Response to Review, the authors responded that they would "qualify this statement to say what is only true of the papers which we found/surveyed in our review." For me, the statement still sounded quite general, not specific to any particular papers.

We have clarified our statement by starting the sentence with "In these models…"

Lines 364-367, "Heterogeneity in carrying capacities increases the likelihood of sustainable harvesters extracting from a resource with a large capacity, which they can maintain at high levels, eventually convincing neighboring nodes to imitate their strategy (Barfuss et al., 2017)." What prevented the non-sustainable harvesters from going to these large-capacity nodes, maintaining themselves at high levels, and convincing others? Again, is this statement valid only for this model? Is it generalizable?

Non-sustainable harvesters are unable to maintain resources at high levels since they harvest at a higher rate. This has been clarified in the text with clearer reference to the model where it was observed.

Line 495, "…as socio-economic data is often more frequently collected and readily available than environmental data": I must say that I have heard the opposite more often.

This was true 20 years ago, but arguably not any more due to the advent of digital social data (Salathé et al., 2012). We've revised this to say "as social-economic data availability is growing faster than ecological data (and perhaps even environmental data despite growth of publicly available satellite data) on account of the era of digital social data."

Technical corrections
- The sole paragraph in Section 2.2 was too long. Please consider breaking it down.
We broke this single paragraph into two paragraphs.

- Lines 235-236, "…made up of a single dominant behavior, which is highly dependent on the initial proportion of behaviors in a population" – wasn't this just describing a stag-hunt game without saying so?
This is not the same as we're describing population game vs an N player, so the mechanisms show up differently.

- Line 255, "where the environmental state is the proportion of infected individuals,…": it felt a bit odd to call infected individuals an "environmental" state, not a "human" state, in a coupled human-environmental system model.
We interpret the prevalence of infection as part of the environment that humans must function in, in much the same way that one may speak of an 'urban environment' for instance. Behavioural-epidemiological systems are commonly included under the umbrella of human-environment systems.

- Line 269, suggest adding "in the parameter space" after "region" to distinguish it from, say, geographical regions.
We have made this change to the text.

- Line 280, "Whereas…" is not a complete sentence. I believe this was meant to be part of the previous sentence.
We have combined the two sentences.

- Line 327, "ncreasing" is missing "i"
This has been corrected.

- Line 351, it's not clear to me what "these commonalities" refer to here. Please clarify.
We are referring to models that show the effects of social trait complexity and have clarified this in the text.

- Line 562, perhaps add "incorporating" in front of "social biases"
This has been added to the text.

- Please double-check the use of punctuation. At many places throughout, I believe ";" should be used in place "," in front of "however". Alternatively, at these places, you should end the previous sentence with a period, start a new sentence with "However,"
We have corrected this use of punctuation.